# Fe Ions-Doped TiO_2_ Aerogels as Catalysts of Oxygen Reduction Reactions in Alkaline Solutions

**DOI:** 10.3390/ma15238380

**Published:** 2022-11-24

**Authors:** Chen Chu, Jinqiong Tang, Zhiyang Zhao, Yong Kong, Xiaodong Shen

**Affiliations:** 1College of Materials Science and Engineering, Nanjing Tech University, Nanjing 210009, China; 2Jiangsu Collaborative Innovation Center for Advanced Inorganic Function Composites, Nanjing 210009, China

**Keywords:** TiO_2_ aerogel, Fe ion doping, oxygen vacancy, pore structure, catalysis, ORR

## Abstract

Aerogels have interconnected networks and preeminent pore structures. When used as the catalysts for oxygen reduction reaction (ORR), they can facilitate the mass transfer and expose more active sites. Here, we synthesized the Fe-doped titanium oxide-based aerogels (TA/Fes) by the sol–gel method combined with thermal treatment. The specific surface areas of the TA/Fes ranged from 475 to 774 m^2^·g^−1^, and the pore volumes varied from 0.96 to 1.72 cm^3^·g^−1^. The doping effect of the Fe ions and the oxygen vacancies in anatase enhance the electrical conductivity, leading to the low Rct (313.3–828.2 Ω). All samples showed excellent stability (2.0–4.5 mV) and 4e^−^ pathway. The limiting current density of TA/Fe3 reached 5.34 mA·cm^−2^, which was comparable to that of commercial Pt/C. The preparation method is inspiring and the as-prepared aerogel catalysts have potential in promoting the scale of fuel cells.

## 1. Introduction

Fuel cells are the most promising power generation technology due to their high energy conversion efficiency and pollution-free aspect. However, the sluggish ORR (oxygen reduction reaction) of cathode has drastically limited their development [1,2,3]. Pt and its alloys have been commercially used as catalysts of ORR due to their high catalytic activity [4]. Nevertheless, these drawbacks, such as low reserves, high cost, and poor stability, necessitate the search for reliable alternatives [5,6,7,8]. Due to the unfilled d-orbitals, transition metal oxides, carbides, nitrides, and carbonitrides, single-atom catalysts of ORR exhibited outstanding catalytic activity [9,10,11], and have been deemed as the ideal replacement. Among these, titanium oxides have been employed as catalyst supports of ORR for their excellent electrochemical stability.

Titanium oxides form three naturally occurring polymorphic crystalline modifications in the form of the corresponding minerals: brookite with rhombic, anatase, and rutile with a tetragonal crystal lattice [12,13,14]. Rutile is the most thermodynamically stable modification. During heating, anatase and brookite irreversibly transform into rutile, and the stability of the crystalline modification depends on the size of its constituent crystallites [15,16]. Meanwhile, titanium oxides can have nanostructures with different phase compositions and morphologies, in particular as nanoparticles, nanorods, nanowires, nanotubes, and mesoporous structures [17], whereas they perform poorly when used as catalysts [18,19,20,21,22,23,24]. 

There are two approaches to accomplish the enhancement of catalytic performance [25]. The first approach is to enhance electrical conductivity, which can accelerate the electron transfer. Introducing high conductivity materials, oxygen vacancies, and foreign elements is the effective way to achieve this [26,27,28,29]. Optimizing the pore structure is the second approach, which facilitates the mass transfer [30]. Both the synthesis procedure and post-processing can alter the pore structure. However, the pore structures of titanium oxides that have been reported still need a lot of improvements. Additionally, the pore dynamics of titanium oxides may be studied in detail by positron annihilation lifetime spectroscopy (PALS) [31].

Aerogels with high specific surface areas and high pore volumes have interconnected networks. When they are used as catalysts, mass transfer and reaction kinetics of ORR are bound to be improved, so as the catalytic performance. Fe, as a transition metal with unfilled d-orbital electrons (3d^6^4s^2^) and multiple valence states (4+, 3+, 2+), can be used as a catalyst for electrocatalytic reactions. In this work, Fe ions-doped TiO_2_ aerogels (TA/Fes) with high specific surface areas and pore volumes were developed by doping different valence states and different binding states of iron sources. The composition, structure, and ORR catalytic performance of the TA/Fes (Fe ions-doped TiO_2_ aerogels) were investigated comprehensively.

## 2. Material and Methods

### 2.1. Synthesis of the Fe Ions-Doped Titanium Oxide-Based Aerogel Catalysts

Resorcinol-formaldehyde (RF, 99% purity, China National Pharmaceutical Group Corp., Beijing, China) was selected as the carbon source, and tetrabutyl titanate (TBOT, ≥99% purity, Aladdin, Shanghai, China) was the titanium source. For the RF (resorcinol-formaldehyde)/Fe sol, 0.001 mol Fe source, 0.05 mol of resorcinol, 0.2 mol of formaldehyde solution (37 wt·%), 25 mL of ethanol, and 1.8 mL of dilute hydrochloric acid were mixed in the beaker within 15 min. For the TiO_2_ sol, solution A (0.05 mol of tetrabutyl titanate and 40 mL of ethanol) and solution B (40 mL of ethanol, 3.6 mL of deionized water, and 1 mL of nitric acid) were mixed within 10 min. The two sols were then homogenously mixed to carry out the sol–gel process, and the wet gels (RF/Fe/TiO_2_) were prepared. The resulting RF/Fe/TiO_2_ gels were washed with ethanol at 50 °C for 3 days to obtain the alcogels. The alcogels were dried with the technique of supercritical CO_2_ drying (SCD) to prepare the RF/TiO_2_ aerogels. Then, the as-prepared aerogels were thermal-treated at 900 °C and held for 5 h to partially reduce N_2_ flow (100 mL·min^−1^) with a heating rate of 2 °C·min^−1^. The Fe sources were Fe(NO_3_)_3_·9 H_2_O (99% purity, Shanghai Bailingwei, Shanghai, China), C_15_H_21_FeO_6_ (98% purity, Aladdin, Shanghai, China), K_4_Fe(CN)_6_·3 H_2_O (98% purity, Sigma-Aldrich, St. Louis, MO, USA), and Fe_3_O_4_ (98% purity, Aladdin, Shanghai, China), and the as-prepared aerogel catalysts were denoted as TA/Fe1, TA/Fe2, TA/Fe3, and TA/Fe4, respectively. The synthesis process of the TA/Fes is illustrated in Figure 1.

### 2.2. Synthesis of the Aerogel Catalyst Ink

A total of 4 mg of aerogel catalyst was ultrasonically dispersed in a solution containing 1 mL ethanol, 1 mL isopropyl alcohol (IPA, 99.7% purity, Shanghai Lingfeng, Shanghai, China), and 50 µL Nafion (5 wt%, Dupont, Wilmington, DE, USA) in an ice/water bath for 30 min. To prepare the working electrode (WE), 16 µL of catalyst ink was coated on rotating disk electrode (RDE, Φ = 5 mm) and rotating ring-disk electrode (RRDE, Φ = 4 mm). The WE was dried at room temperature. The loadings on the RDE (rotating disk electrode) and RRDE (rotating ring-disk electrode) were 0.159 mg·cm^−2^ and 0.248 mg·cm^−2^, respectively.

### 2.3. Characterization Methods

X-ray diffraction (XRD) data were acquired by a SmartLab apparatus (Rigaku, Tokyo, Japan). N_2_ adsorption/desorption isotherms were obtained at 77 K using a BELSORP-Minix (MicrotracBEL, Osaka, Japan) apparatus. The Brunaur–Emmett–Teller (BET) method was used to calculate the specific surface area. The non-local density functional theory (NLDFT) model was adopted to calculate the pore size distribution. The microstructure was characterized by field-emission scanning electron microscopy (FESEM) using a JSM-7600F microscope (JEOL, Tokyo, Japan). The chemical valence states were characterized by X-ray photoelectron spectroscopy (XPS) using a Nexsa spectroscope (Scientek, Delta, BC, Canada). The data were calibrated by carbon with binding energy of 284.8 eV.

### 2.4. Electrochemical Performance Measurements

The ORR performances of the aerogel catalysts were investigated on the RRDE-3A device (ALS, Japan) connected to the CS2350H electrochemical workstation (CORRTEST, Wuhan, China) by a standard 3-electrode system in a 0.1 M KOH electrolyte. Carbon rod (Φ = 6 mm) and 3.5 M Ag/AgCl were used as the counter electrode (CE) and reference electrode (RE), respectively. All potentials obtained from the WE were normalized to reversible hydrogen electrode (RHE) according to the Nernst equation (Equation (1)).
(1)$$E_{RHE}=E_{Ag/AgCl}+0.2046+0.0591\times PH$$

The potential range of all measurements was from −0.9 V to 0.2 V vs. Ag/AgCl. Cyclic voltammetry (CV) test was conducted at a scan rate of 100 mV·s^−1^ without rotating, and the CV curves were obtained in N_2_ or O_2_ for 30 min. Linear weep voltammetry (LSV) test was carried out at a scan rate of 10 mV·s^−1^ and a fixed rotation rate of 1600 rpm. Accelerated durability test (ADT) was used to evaluate the stability of the catalyst, which was assessed by the difference in half-wave potential(ΔE_1/2_) of the LSV curve. Electrochemical impedance spectroscopy (EIS) was performed in a frequency range of 10^6^–0.01 Hz at an open circuit in an oxygen-saturated 0.1 M KOH with an AC amplitude of 5 mV rms. Rotating ring-disk electrode (RRDE) test for ORR was conducted with a sweep rate of 10 mV·s^−1^ and keeping a constant ring potential of 0.3 V vs. Ag/AgCl. The peroxide yields (*HO*_2_^−^%) and transfer number (n) were calculated by the following Equations (2) and (3).
(2)$${HO}_{2}^{-}\%=\frac{200J_{r}}{N(J_{d}+\frac{J_{r}}{N})}$$
(3)$$n=\frac{4J_{d}}{J_{d}+\frac{J_{r}}{N}}$$
where *J_d_* is the disk current density; *J_r_* is the ring current density; *N* represents the current collection efficiency of the Pt ring, and the value of it is 0.414, which is determined by the reduction in K_3_Fe(CN)_6_.

## 3. Results and Discussion

### 3.1. Materials Characterizations

The photographs of the titanium oxide aerogels doped by different Fe sources after heat treatment are shown in Figure 2. The color on the surface of the TA/Fe1 and TA/Fe2 distinctly exhibited the blue color, while the surface of the TA/Fe3 and TA/Fe4 with a very minimal blue and light brown were generally black. The color change indicated the transformation of the crystalline phase.

XRD (X-ray diffraction) analysis was applied to further confirm the phase change of the aerogels. As Figure 3 shows, both anatase (PDF NO.89-4921) and rutile (PDF NO.71-0650) existed in the TA/Fe2, which suggested the appearance of the mixed crystalline phase. In the TA/Fe1, TA/Fe3, and TA/Fe4, only the anatase was formed.

N_2_ adsorption was used to evaluate the pore structure of the as-prepared aerogel catalysts (Figure 4). As Figure 4a shows, all samples exhibited type IV isotherms with H3 hysteresis loops according to the IUPAC classification, which suggested the existence of narrow mesopores and macropores in the materials. The adsorption within the low relative pressure region (below P/P_0_ = 0.05) was due to the micropores. After thermal treatment at 900 °C, the organic constituent (i.e., resorcinol-formaldehyde) was decomposed into carbon, which generated the micropores. The medium relative pressure region (P/P_0_ = 0.05~0.80) implied the magnitude of the specific surface area. The TA/Fe1 had the highest specific surface area of 774 m^2^·g^−1^. There was no platform in the isotherms close to P/P_0_ = 1.0, indicating that all samples had macropores. The pore volume and median pore size of the TA/Fe3 are 1.72 cm^3^·g^−1^ and 51.6 nm, respectively. The specific surface area of the TA/Fe3 is 543 m^2^·g^−1^. The TA/Fe2 are mainly microporous and mesoporous. Under this doping condition, the specific surface area and pore volume of the TA/Fe2 are 475 m^2^·g^−1^ and 0.96 cm^3^·g^−1^, respectively. At this time, the pore volume is at the minimum and the median pore size is only 22.9 nm. These analyses exhibited that the TA/Fes possessed the hierarchical micro–meso–macropores, which were in line with the pore size distribution curves presented in Figure 4b. The pore structure data are shown in Table 1. It should be highly emphasized that the molecule size of the O_2_ and H_2_O were 0.346 nm and 40 nm, respectively, which suggested that the hierarchical pore structure would facilitate the catalysis.

In order to observe the microstructure of the aerogel catalysts, the FESEM characterization was used. The as-prepared samples maintained the typical colloidal structure of aerogels with numerous nanoparticles and nanopores after heat treatment (Figure 5). After the doping of the C_15_H_21_FeO_6_ (Figure 5b), the inner wall of the pore adhered to some cubic particles, which implied that the doping effect could influence the microstructure. Moreover, its pore structure remained intact due to the low pore volume and small median pore diameter. In general, the interconnected structure and porosity of these aerogel catalysts favored the mass transfer, which was conducive to the catalytic process.

XPS (X-ray photoelectron spectroscopy) was used to study the valence state and the surface state. Figure 6 presents the XPS spectra of Ti 2p. The Ti 2p spectra were divided into four peaks after fitting. Among them, ~459.0 eV and ~465.0 eV were assigned to the Ti 2p_3/2_ and Ti 2p_1/2_ of Ti^4+^, respectively, while the two other peaks located at ~460.0 and ~462.0 eV were attributed to the Ti 2p_3/2_ and Ti 2p_1/2_ of Ti^3+^, respectively, which was in accordance with the former literature. The shift to the lower binding energy indicated that part of the Ti^4+^ was reduced to Ti^3+^, and the oxygen vacancies were generated for the electrostatic balance. Although the XRD characterization did not show the crystalline phase of suboxides, the XPS spectra suggested the formation of Ti^3+^. The O 1s spectra are shown in Figure 7, which are deconvoluted into three peaks. The binding energy of ~530.3 eV was ascribed to the O_L_–Ti bonding, which was associated with the lattice oxygen. The peaks located at ~532.3 eV and ~533.7 eV were considered as O_ads_–Ti bonding and O–C bonding, respectively. The ratio of the peak area of O_ads_–Ti and O–C to the total peak area illustrated the adsorption ability of oxygen and a higher value corresponded to a better adsorption. As Table 2 shows, by calculating, we found that the ratio followed this rule: TA/Fe1 < TA/Fe2 < TA/Fe4 and TA/Fe3, so as the adsorption ability. The aerogel catalysts doped by K_4_Fe(CN)_6_·3H_2_O and Fe_3_O_4_ had the highest adsorption ability, which may be a result of the high pore volume and high median pore diameter, and the doping of Fe(NO_3_)_3_·9H_2_O and C_15_H_21_FeO_6_ exhibited lower adsorption abilities due to the small median pore diameter and pore volume.

### 3.2. Electrochemical Performances

CV (cyclic voltammetry) was first used to evaluate the ORR catalytic performance of the materials. The titanium oxide-based aerogels doped by different Fe sources are shown in Figure 8. It can be found that no cathodic peak concerning oxygen reduction was found in the nitrogen test atmosphere (Figure 8a). On the contrary, the cathodic peaks were observed in all aerogel catalysts under the oxygen test atmosphere, confirming the reaction of oxygen reduction (Figure 8b). The high peak current and high peak potential possess better ORR catalytic activities. The TA/Fe4 presented the largest peak current of 1.41 mA·cm^−2^ and the TA/Fe2 had the smallest peak current of 0.74 mA·cm^−2^. However, the latter exhibited a higher peak potential of 0.68 V.

A further characterization of the ORR catalytic performance was carried out using LSV (linear weep voltammetry), as shown in Figure 9. The TA/Fe3 had the highest limiting current density (J_limit_) of 5.34 mA·cm^−2^, which was close to the commercial Pt/C (5.55 mA·cm^−2^), while the TA/Fe2 showed the lowest current density of 4.51 mA·cm^−2^. Still, the values of J_limit_ were higher than their counterparts due to the high specific surface area and large pore volume, which was conducive to the exposure of active sites and mass transfer. Anatase is octahedrally distorted due to octahedral mismatch, which also leads to the occurrence of oxygen vacancies and structural defects. Compared to anatase, the rutile crystalline structure with fewer defects is more stable. Therefore, the electrical conductivity of anatase is better than that of rutile. In the TA/Fe2, two crystal structures appeared simultaneously, i.e., mixed crystals. The formation of a large amount of rutile reduced the electrical conductivity of the material, thus hindering the electron transfer. The stability was evaluated by comparing the difference in the half-wave potential (ΔE_1/2_) of the LSV curves before and after 1000 CV cycles. The ORR catalytic performance data are shown in Table 3. It was not difficult to find that all samples exhibited high electrochemical stability.

Tafel slope was used to describe the reaction kinetics of ORR, as shown in Figure 10. A smaller Tafel slope indicated a smaller transfer resistance; hence, the reaction occurred easily. From the data in the figure, it can be seen that the performance of the organic iron-doped sample (Fe2) is inferior to that of the inorganic iron sources. This could be explained by two reasons. The first was due to the lowest pore volume and specific surface area, which hindered the mass transfer. The second was the deterioration of the electrical conductivity due to the formation of large amounts of rutile.

The transfer number (n) and hydrogen peroxide yield (HO_2_^−^) were evaluated by potentiostatic polarization, as shown in Figure 11. The TA/Fe1 had the lowest HO_2_^−^ yield in the potential range of 0.2–0.8 V; however, the TA/Fe2 were higher than the others. Nevertheless, all samples showed the 4e^−^ transfer pathway.

The EIS (electrochemical impedance spectroscopy) test was used to evaluate the impedance during the electrochemical reaction, as shown in Figure 12. From high frequency to low frequency, the electrons were transported from the surface to the inside of the catalysts. A corrosion medium/corrosion product/metal fitting circuit was used. The equivalent-circuit model had five parameters including Rs, R1, C1, Q1, and Rct, which referred to the solution impedance, the corrosion product coating impedance, the electrolyte/corrosion product coating interface capacitor, the corrosion products/metal interface capacitor, and the charge transfer impedance, respectively. The fitting data are provided in Table 4. The Rs is related to the electrolyte and the geometry of the test cell, rather than the characteristics of the catalysts. Therefore, the Rs is generally used for the IR compensation. Rct is the crucial parameter concerned with the resistance of the electrochemical reaction, with low Rct favoring the electron transfer. The Rct of PRTA/Fe1-900N, PRTA/Fe2-900N, PRTA/Fe3-900N, and PRTA/Fe4-900N were 444.4 Ω, 828.2 Ω, 497.8 Ω, and 313.3 Ω, respectively. Since rutile in the TA/Fe2 is the stable form of TiO_2_ compared to anatase, which leads to the differences in their electronic structures and forbidden bandwidths, the differences in electrical conductivity and catalytic activity appeared. The Rct values of the as-prepared aerogel catalysts were smaller than the research that has been reported on TiO_2_, which also illustrated that the doping influenced the electrical conductivity by introducing an impurity level in the bandgap and thus enhancing the electron transfer. In addition, a larger pore volume and a larger median pore size were more favorable to the mass transfer and lowered the reaction impedance, which improved the catalytic activity of ORR.

### 3.3. Influence of Oxygen Vacancies, Doping Effect, and Pore Structure

Fe-doped titanium oxide-based aerogel catalysts had better electrical conductivity and ORR catalytic activity due to the impurity levels of oxygen vacancies and Fe ions. In addition, the ORR catalytic performance of the TA/Fe3 were better than those of the undoped materials (initial potential (0.80 V), half wave potential (0.71 V), highest limiting current density (4.84 mA·cm^−2^), stability of electrochemistry (1.2 mV), and Tafel slope (70 mV·dec^−1^)) [32]. The formation of oxygen vacancies, which was proved by the XPS characterization, would serve as donors. The Fe ions in the materials would serve as either donors or acceptors, depending on the form of the metal atoms in the lattice. When a metal atom replaced a lattice atom, a donor level was created; if a metal atom formed as interstitial, an acceptor level appeared. Either way, the bandgap would be narrowed which improved the electrical conductivity, thus lowering the electron transfer resistance. What is more, the oxygen adsorption energy barrier would be reduced due to the formation of oxygen vacancies, changing the oxygen-related descriptors [11,28]. Moreover, the O_2_ could be easily absorbed by the oxygen vacancies, improving the reaction kinetics. Moreover, since the molecule size of H_2_O is 40 nm and for O_2_ is 0.345 nm, it is important to ensure that each reactant can reach the active sites easily. The higher pore volume and the median pore size close to the molecule size of H_2_O would improve the catalytic performance. Therefore, micropores facilitated the transport of electrons and gas molecules, and meso- macropores facilitated rapid mass transfer (Figure 13).

## 4. Conclusions

A novel Fe sources-doped titanium oxide-based aerogel catalyst for ORR was successfully prepared. The as-prepared aerogel catalyst had a high specific surface area and a large pore volume, especially in the TA/Fe3 and TA/Fe4 aerogels. The excellent pore structure facilitated the exposure of active sites. The relationship between the median pore size and the catalytic performance of ORR was discovered. The median pore size close to the molecular size of H_2_O was conducive to the mass transfer of reactants. Hence, it was not enough to barely emphasize the increase in the specific surface area, as the high pore volume combined with a suitable median pore size seemed to be more important. Moreover, the crystalline structure of titanium oxide had a great influence on the conductivity. When a large amount of rutile was generated (TA/Fe2), its conductivity was reduced, thus hindering the electron transfer and slowing the reaction kinetics of ORR. The oxygen vacancies in anatase and the doping with Fe ions increased the electrical conductivity by generating impurity energy levels. This synthesis method can be extended to other transition metal catalysts to improve the mass transfer and the kinetics of catalysts.

## Figures and Tables

**Figure 1 materials-15-08380-f001:**
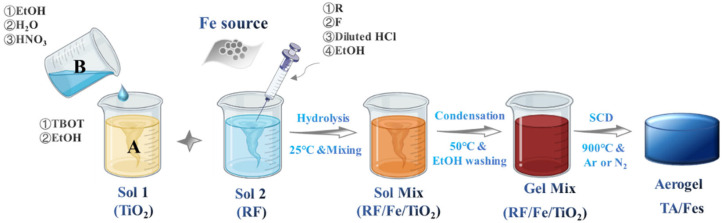
The preparation of titanium oxide-based aerogel catalysts doped by four types of Fe sources.

**Figure 2 materials-15-08380-f002:**
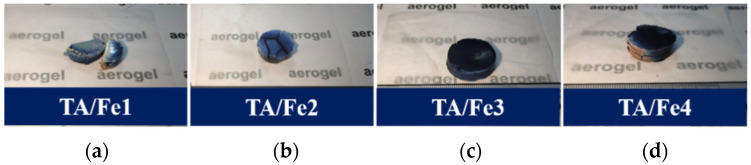
Photographs of TA/Fes: (**a**) doped by Fe(NO_3_)_3_·9 H_2_O, (**b**) doped by C_15_H_21_FeO_6_, (**c**) doped by K_4_Fe(CN)_6_·3H_2_O, and (**d**) doped by Fe_3_O_4_.

**Figure 3 materials-15-08380-f003:**
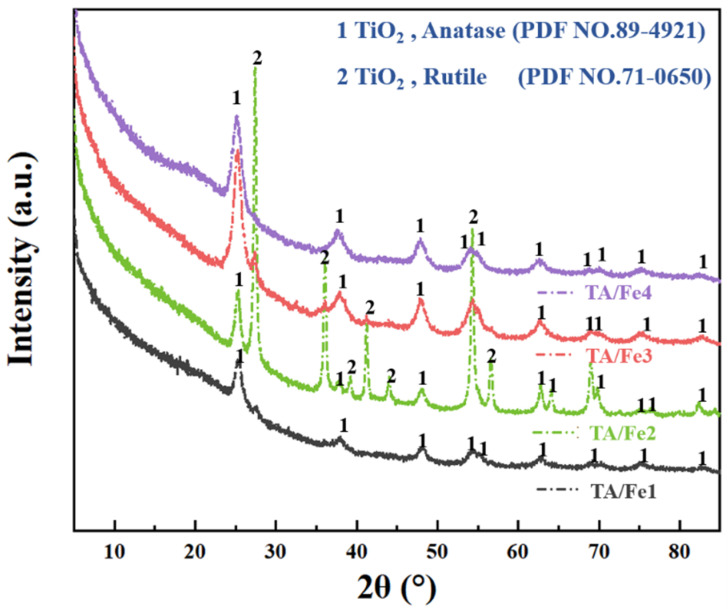
XRD patterns of the TA/Fe1, TA/Fe2, TA/Fe3, and TA/Fe4.

**Figure 4 materials-15-08380-f004:**
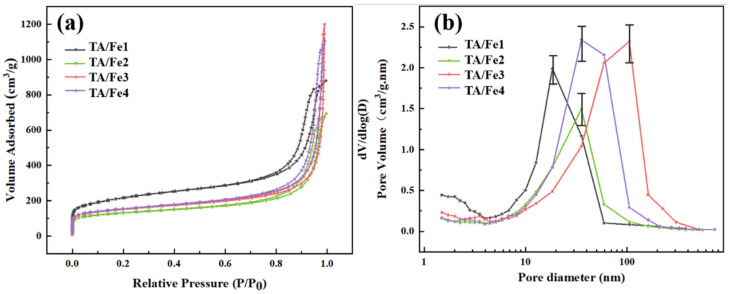
(**a**) N_2_ adsorption/desorption curves and (**b**) pore size distribution of the TA/Fe1, TA/Fe2, TA/Fe3, and TA/Fe4.

**Figure 5 materials-15-08380-f005:**
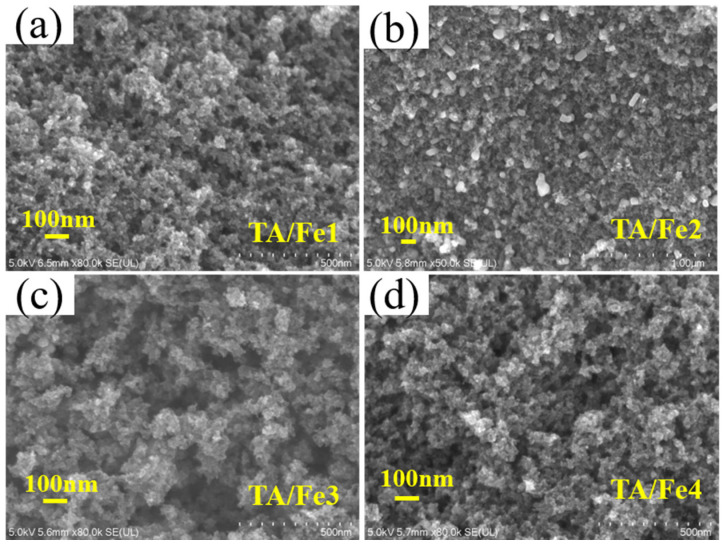
FESEM images of the Fe-doped titanium oxide-based aerogel catalysts.

**Figure 6 materials-15-08380-f006:**
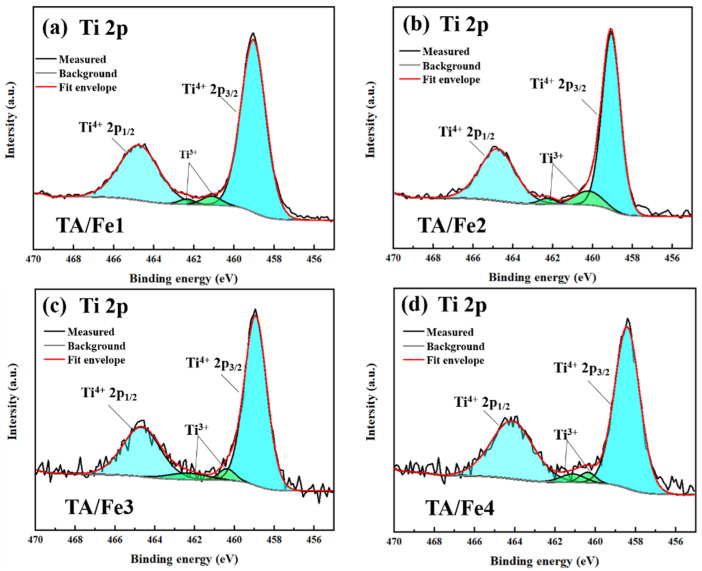
Ti 2p XPS spectra of the titanium oxide-based aerogel catalysts doped by different Fe sources: (**a**) TA/Fe1, (**b**) TA/Fe2, (**c**) TA/Fe3, (**d**) TA/Fe4.

**Figure 7 materials-15-08380-f007:**
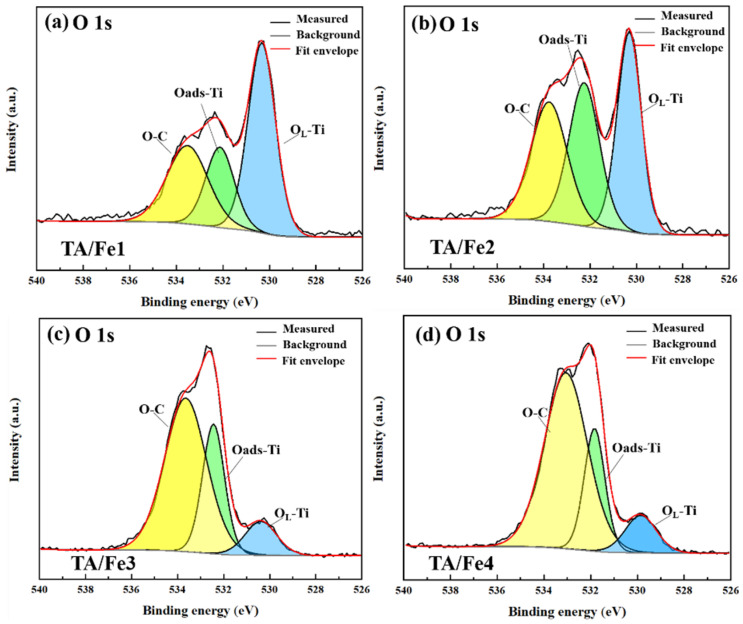
O 1s XPS spectra of the titanium oxide-based aerogel catalysts doped by different Fe sources: (**a**) TA/Fe1, (**b**) TA/Fe2, (**c**) TA/Fe3, (**d**) TA/Fe4.

**Figure 8 materials-15-08380-f008:**
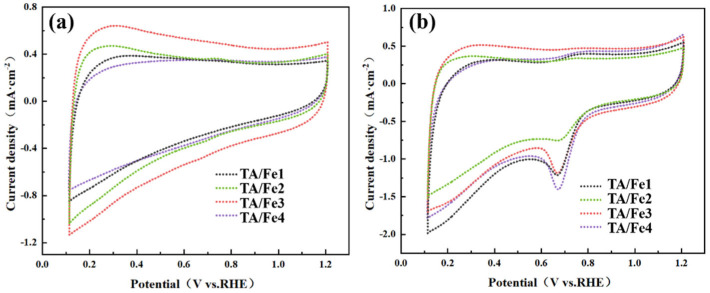
CV curves of the titanium oxide-based aerogel catalysts doped by different Fe sources: (**a**) N_2_ test condition, (**b**) O_2_ test condition.

**Figure 9 materials-15-08380-f009:**
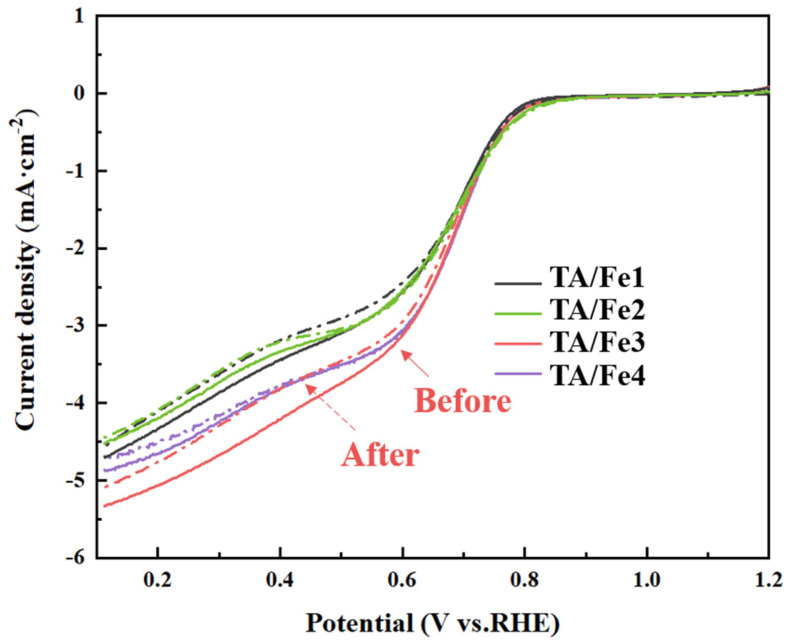
LSV curves and stability of the titanium oxide-based aerogel catalysts doped by different Fe sources.

**Figure 10 materials-15-08380-f010:**
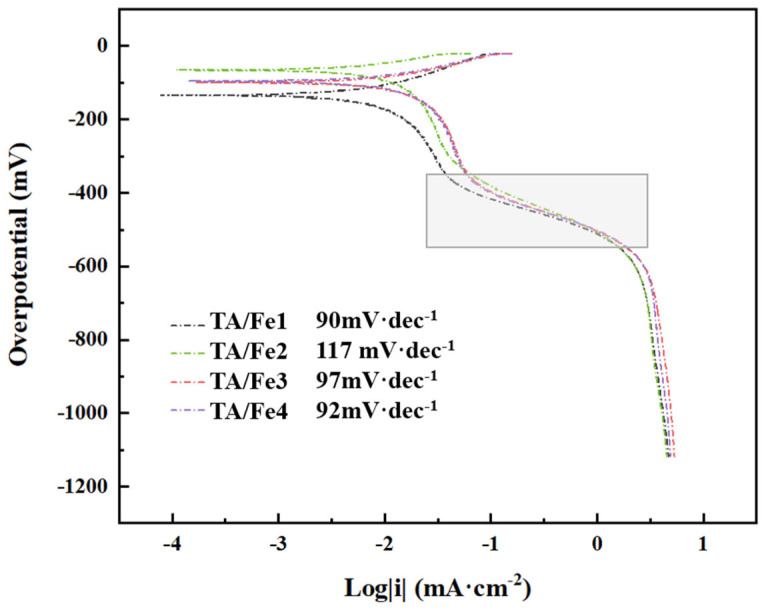
Tafel curves and stability of the titanium oxide-based aerogel catalysts doped by different Fe sources.

**Figure 11 materials-15-08380-f011:**
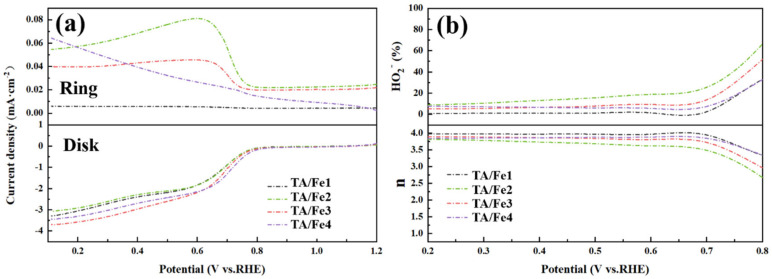
(**a**) Potentiostatic polarization curves and (**b**) hydrogen peroxide yields and transfer number of the as-prepared aerogel catalysts (at the constant voltage of 1.3 V vs. RHE).

**Figure 12 materials-15-08380-f012:**
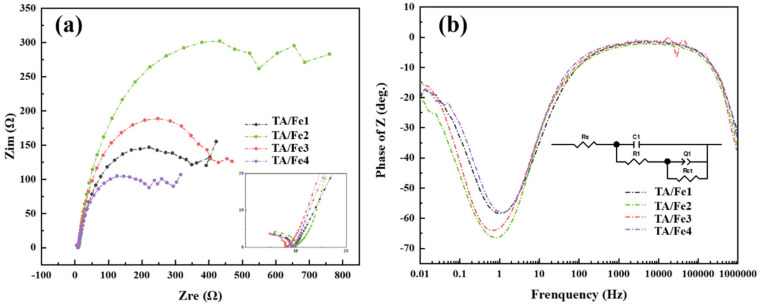
(**a**) Nyquist plots and (**b**) Bode diagrams of the titanium oxide-based aerogel catalysts doped by different Fe sources.

**Figure 13 materials-15-08380-f013:**
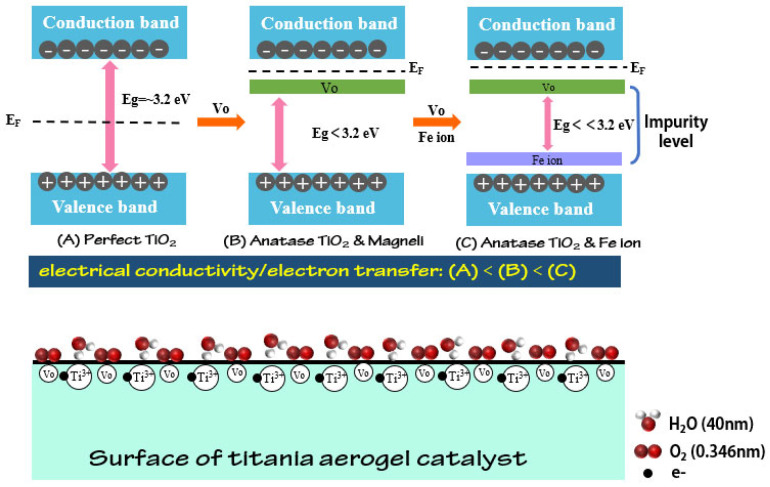
The influence mechanism of the oxygen vacancies, doping effect, and pore structure.

**Table 1 materials-15-08380-t001:** Pore structure parameters of the as-prepared aerogel catalysts.

Sample	Specific Surface Area (m^2^·g^−1^)	Pore Volume (cm^3^·g^−1^)	Median Pore Diameter (nm)
TA/Fe1	774 ± 77.01	1.22 ± 0.02	16.1 ± 0.81
TA/Fe2	475 ± 29.74	0.96 ± 0.01	22.9 ± 1.80
TA/Fe3	543 ± 50.66	1.72 ± 0.06	51.6 ± 5.87
TA/Fe4	551 ± 69.13	1.58 ± 0.04	28.1 ± 3.09

**Table 2 materials-15-08380-t002:** XPS peak areas of O 1s spectra.

Sample	Peak Area of O_L_–Ti	Peak Area of O_ads_–Ti	Peak Area of C–O	Ratio
TA/Fe1	12,024.92 ± 112.4	5269.99 ± 64.23	7390.67 ± 76.54	0.51
TA/Fe2	6587.58 ± 82.5	6017.46 ± 68.12	5894.03 ± 60.03	0.64
TA/Fe3	4157.14 ± 64.6	10,971.72 ± 106.3	24,772.02 ± 232.35	0.89
TA/Fe4	3881.96 ± 58.2	8421.86 ± 91.02	25,195.61 ± 261.62	0.89

**Table 3 materials-15-08380-t003:** ORR catalytic performance data of the as-prepared aerogels.

Sample	Epeak	Jpeak	Eonset (V)	E_1/2_ (V)	|J_limit_| (mA·cm^−2^)	|Tafel slope| (mV·dec^−1^)	Stability (mV)
TA/Fe1	0.67	1.21	0.81	0.70	4.73	89	2.0
TA/Fe2	0.67	0.74	0.80	0.71	4.51	117	4.5
TA/Fe3	0.67	1.18	0.78	0.71	5.34	97	4.5
TA/Fe4	0.68	1.41	0.78	0.71	4.88	92	2.5

**Table 4 materials-15-08380-t004:** Electrochemical impedance data of as-prepared aerogel catalysts.

Sample	Rs (Ω·cm^2^)	C1 (nF/cm^2^)	R1 (Ω·cm^2^)	Q1-Q (nF/cm^2^)	Q1-n	Rct (Ω·cm^2^)
TA/Fe1	2.53	25.4	6.955	0.78	0.788	444.4
TA/Fe2	1.88	21.1	8.225	0.98	0.834	828.2
TA/Fe3	1.25	22.9	7.191	1.41	0.841	497.8
TA/Fe4	2.57	28.3	6.656	1.13	0.812	313.3

## Data Availability

Not applicable.

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
