# Peer review of "Fe Ions-Doped TiO2 Aerogels as Catalysts of Oxygen Reduction Reactions in Alkaline Solutions"

_materials, 2022, doi:10.3390/ma15238380_

Round 1

Reviewer 1 Report

The manuscript by Shen et al. demonstrated Fe ion doped TiO2 aerogels for catalysis applications. The study is systematic, and the manuscript is well organized. The only concern I have on this study is about the doping of Fe ions. In this study, the doping rate or concentration of Fe ions have not been characterized yet, though XPS methods are used. Actually, the doping rate could be a critical factor in affecting the electrical conductivity (this was also absent in the study) and microstructures. For example, in the Figure 2, the authors claimed there is a transition between samples made by different iron sources and verified this by XRD. However, why there is a transition and what is the cause remains unclear. I am still confused about the motivation of using different iron sources, are this used for modulating the doping rate? I would suggest the authors to include detailed characterization of doping rates for this study. Other minor thing is that the mechanism part is too simple and contains too much vague information. At current stage, I would suggest a minor revision for this manuscript.

Reviewer 2 Report

The present manuscript is interesting and clearly written. It can be accepted for publication, after considering the following remarks:

11)     Title is incomplete.

22)     The use of English language (for example: excessive use of “the”, missing nouns, typos) should be checked.

33)     All abbreviations should be explained when they first appear in the main manuscript.

44)     Section 2. The origin and purity characteristics of all materials used should be given, if available.

55)      The authors should comment on the reproducibility of the synthesis process.

66)     Where there checked other doping levels?

77)     A discussion of the results in comparison with the respective undoped material would be of interest.

Reviewer 3 Report

Referee report on “Preparation of Fe ions doped TiO2 aerogels as catalyst for oxygen reduction reaction in alkaline" by Chen Chu et al.

Although this topic is of some interest, this manuscript in its present form cannot be recommended for publication and requires at least some improvement.

1.     Although the introduction is well written, several points are either not disclosed or not completely accurate:  a) Iron can be in several charge states (Fe2+, Fe3+ etc). The charge state determines its local environment, for example, the presence of compensating vacancies. b) It is important to note that in addition to the traditional methods, new methods for studying pores have been recently developed, which makes it possible to study their dynamics in detail. See recent MDPI paper:

Klym, H.; et al. Positron Annihilation Lifetime Spectroscopy Insight on Free Volume Conversion of Nanostructured MgAl2O4 Ceramics. Nanomaterials 202111, 3373. https://doi.org/10.3390/nano11123373

2.    More new information about TiO2. In order for the article to be of interest to a wider readership, more information about titanium oxide would be useful to include. A lot of recent studies have been published in MDPI journals. It is known that TiO2 can be in different crystalline modifications, and the nanostructure is also very diverse, both in structure and size. Few of them are below:

Serga, V.; et al Crystals 202111, 431. https://doi.org/10.3390/cryst11040431

Tsebriienko, T.; et al. Crystals 202111, 794. https://doi.org/10.3390/cryst11070794

3.     Formulas 1 and 2 need revision to meet journal standards.

4.     Figure 4 (b) requires an error bar and corresponding discussion in the text.

5.     Data in Table 1 also requires an error bar and corresponding discussion in the text.

6.     Fig. 5. How stable are these images over time? Does aging occur?

7.     Data in Table 2 also requires an error bars.

8.     Line 205. Give more explanation about mixed crystals. Can this be proved or confirmed by structural methods (Raman, for example?)

9.     Reference list.  Reference [4] needs to be checked (volume or page number are absent). Also references 7, 8, 11, 12, 15, 19, 21 need to be checked, too.

 In general, the manuscript is interesting and can be recommended for publication after constructive reflection on the above comments.

Round 2

Reviewer 3 Report

The Authors have significantly improved their original manuscript

Which now can be recommended for publication